# The Pathological Mechanisms of Hearing Loss Caused by *KCNQ1* and *KCNQ4* Variants

**DOI:** 10.3390/biomedicines10092254

**Published:** 2022-09-12

**Authors:** Kazuaki Homma

**Affiliations:** 1Department of Otolaryngology-Head and Neck Surgery, Feinberg School of Medicine, Northwestern University, Chicago, IL 60611, USA; k-homma@northwestern.edu; Tel.: +1-312-503-5344; 2The Hugh Knowles Center for Clinical and Basic Science in Hearing and Its Disorders, Northwestern University, Evanston, IL 60608, USA

**Keywords:** *KCNQ1*, *KCNQ4*, Kv7.1, Kv7.4, hearing loss, DFNA2A, Jervell and Lange–Nielsen syndrome, long QT syndrome, Romano–Ward syndrome

## Abstract

Deafness-associated genes *KCNQ1* (also associated with heart diseases) and *KCNQ4* (only associated with hearing loss) encode the homotetrameric voltage-gated potassium ion channels Kv7.1 and Kv7.4, respectively. To date, over 700 *KCNQ1* and over 70 *KCNQ4* variants have been identified in patients. The vast majority of these variants are inherited dominantly, and their pathogenicity is often explained by dominant-negative inhibition or haploinsufficiency. Our recent study unexpectedly identified cell-death-inducing cytotoxicity in several Kv7.1 and Kv7.4 variants. Elucidation of this cytotoxicity mechanism and identification of its modifiers (drugs) have great potential for aiding the development of a novel pharmacological strategy against many pathogenic *KCNQ* variants. The purpose of this review is to disseminate this emerging pathological role of Kv7 variants and to underscore the importance of experimentally characterizing disease-associated variants.

## 1. Introduction

The human genome encodes 40 voltage-dependent potassium ion (K^+^) channels that belong to 12 subfamilies (Kv1–Kv12). The voltage-dependent activities of Kv channels play critical roles in controlling the electrophysiological properties of cells and maintaining the ion homeostasis. All Kv channels possess six transmembrane (TM) segments (S1–S6). The first four TM segments (S1–S4) constitute the voltage-sensing domain, while the last two, which flank a channel pore loop (P-loop), constitute the pore-forming domain (S5-PH-S6) (Figure 1). Tetramerization is essential for completing the K^+^-selective channel pore in all Kv channel types. Two Kv channels, Kv7.1 and Kv7.4, are essential for normal operation of the inner ear [1,2].

Kv7.1 is encoded by *KCNQ1*. It is abundantly expressed in the heart and is crucial for normal repolarization of cardiomyocytes. Mutations in Kv7.1 underlie two forms of the long QT syndrome (LQTS), i.e., the Romano–Ward syndrome (RWS) [3] and Jervell and Lange–Nielsen syndrome (JLNS) [4]. Cardiac symptoms in JLNS are typically more severe than those in RWS, and most patients with JLNS also suffer from congenital hearing loss [4]. RWS is inherited dominantly, with some exceptions [5,6], whereas JLNS is inherited recessively. In the cochlea, Kv7.1 is expressed in the marginal cells of the stria vascularis (SV) [4]. Kv7.1 is thought to mediate the secretion of K^+^ into the endolymph and the establishment of the endocochlear potential (EP) [7].

Kv7.4 is encoded by *KCNQ4*. In the cochlea, two types of sound-sensing cells, inner hair cells (IHCs) and outer hair cells (OHCs), are housed in the organ of Corti, and their apical surfaces exposed to the endolymph. Kv7.4 is abundantly expressed in OHCs, but it is also expressed in IHCs and the spiral ganglion neurons (SGNs) [8,9]. The K^+^ conductance mediated by Kv7.4 contributes to the establishment of a normal resting membrane potential and is crucial for repolarization of the cells after sound-elicited cell depolarization. The large Kv7.4-mediated conductance in OHCs also contributes to the reduction of the membrane time constant so that the receptor-potential-induced mechanical response of OHCs, i.e., electromotility [10,11], can respond to sound stimuli at high frequencies [12]. K^+^ that flows into OHCs from the high-K^+^-containing endolymph via the stereocilia is thought to be extruded by Kv7.4 from the base of OHCs to the perilymph. This Kv7.4-mediated extrusion of K^+^ is believed to be crucial for maintaining the intracellular ionic homeostasis and, thus, for OHC maintenance. Mutations in Kv7.4 are responsible for dominantly inherited progressive nonsyndromic hearing loss, DFNA2A [13].

## 2. The Pathologies of JLNS and DFNA2A Hearing Loss

Hearing loss in JLNS patients is congenital, bilateral, and profound (OMIN: 220400). To date, 768 *KCNQ1* variants have been reported, ~5% of which are JLNS-associated (the Human Gene Mutation Database, HGMD) [14] (Table 1). The ionic composition of endolymph is unique among extracellular fluids in that it contains high K^+^ (~160 mM) but low Na^+^ (~1 mM) and Ca^2+^ (~20 µM) [15]. The high K^+^ in endolymph is due to the secretion of K^+^ from the marginal cells of the SV, which is powered by ATP-dependent Na^+^/K^+^ pumps, a Na^+^/K^+^/2Cl^−^ cotransporter, and other ion transporters. K^+^ transport from the marginal cells to the endolymph is electrogenic, resulting in positive EP (+80 to +100 mV) [16]. This very positive EP is a main driving force for sensory mechanotransduction by both hair cells (conversion of sound-induced mechanical vibrations sensed by stereocilia into changes in the receptor potential). Kv7.1, together with its ancillary protein, KCNE1, is thought to mediate K^+^ transport from the marginal cells to the endolymph. Thus, JLNS-causing *KCNQ1* variants are presumed to affect EP by impairing the K^+^ conductance of Kv7.1. In fact, *Kcnq1^−^*^/*−*^ mice lose EP and show atrophy of the SV, reduction of the endolymphatic compartment, and degeneration of the organ of Corti and SGNs [17,18,19,20].

Hearing loss in DFNA2A patients is typically progressive and more prominent at higher frequencies (middle and low frequencies are also affected later in life) (OMIN: 600101). The severity of DFNA2A hearing loss and the rate of progression vary among *KCNQ4* variants. To date, 76 *KCNQ4* variants have been reported in HGMD (Table 2). Given the multiple aforementioned roles of Kv7.4 in OHCs, it is anticipated that DFNA2A-causing *KCNQ4* variants impair normal cochlear operation by primarily affecting OHCs. *Kcnq4^−^*^/*−*^, *Kcnq4^G286S^*^/*+*^, and *Kcnq4^G286S^*^/*G286S*^ (p.G285S in humans) mice recapitulate progressive hearing loss that is, indeed, accompanied by OHC dysfunction and degeneration [52]. Degeneration of IHCs and SGNs at later postnatal ages was also found [53].

The presence of multiple JLNS-/DFNA2A-associated variants found in *KCNQ1*/*KCNQ4* and identification of hearing phenotypes in *Kcnq1* and *Kcnq4* mouse models compellingly establish the essentiality of these Kv7 channels in hearing.

## 3. The Pathological Roles of Kv7.1 and Kv7.4 Variants and Ongoing Pharmacological Strategies

A dominant-negative inhibition (inhibition of the Kv7 channel function or membrane targeting by having a mutated subunit in the tetramer complex) often underlies the observed dominant inheritance of Kv7.1 and Kv7.4 variants associated with RWS and DFNA2A. Since tetramerization of Kv7 channels is mediated by the C-terminal cytosolic helices C and D (HC and HD in Figure 1) [95,96,97,98,99,100,101,102,103], haploinsufficiency is presumed to account for the dominant inheritance of frame-shifted or nonsense Kv7 variants that lack the C-terminal tetramerization region. In any case, the functional consequence of Kv7 variants would be the reduction of overall K^+^ channel activity. Hence, pharmacological augmentation of reduced residual Kv7 channel activity by channel openers is considered clinically effective and actively pursued [54,102,103,104,105,106,107,108,109,110,111,112,113,114].

## 4. Observations against a Haploinsufficiency-Based Pathological Mechanism

Kv7.1 variants associated with JLNS are inherited recessively. In other words, individuals heterozygous for JLNS-associated Kv7.1 variants suffer from neither LQTS nor hearing loss. This suggests that one functional *KCNQ1* allele is sufficient for maintaining normal cardiac and auditory functions, thus arguing against a haploinsufficiency-based pathological mechanism. It is perplexing that, among truncated Kv7.1 variants lacking the C-terminal tetramerization region, some are inherited dominantly, while the others are inherited recessively.

Multiple variants truncating the C-terminal tetramerization region are also identified in *KCNQ4*. If haploinsufficiency accounted for dominant inheritance of these truncated Kv7.4 variants, they should all result in a similar and a relatively mild DFNA2A phenotype compared to those that exert a dominant-negative inhibitory effect on the wild-type Kv7.4 (Kv7.4^WT^) subunit. The slowly progressive hearing loss found in patients with heterozygous Kv7.4^Q71Sfs^ (c.211delC) [90] is in line with such a view. However, Kv7.4^W242X^ (c.725G>A), which completely lacks the channel pore-forming transmembrane domain and the following C-terminal tetramerization region (Figure 1), is associated with severe to profound hearing loss [64]. The presence of recessively inherited truncated Kv7.4 variant, Kv7.4^A349Pfs^ (c.1044_1051del8) [92], is also incompatible with a haploinsufficiency-based pathological mechanism. Incidentally, *Kcnq4^+^*^/*−*^ heterozygous mice do not suffer from hearing loss [52], suggesting that one functional *KCNQ4* allele is sufficient for maintaining normal auditory function, at least in mice.

## 5. Identification of Cell Death-Inducing Cytotoxicity in Truncated Kv7.1 and Kv7.4 Variants

In order to elucidate the pathogenic roles of truncated Kv7 variants, we first tried to confirm the absence of K^+^ channel activity for three deafness-associated Kv7.4 variants that completely lack the C-terminal tetramerization region, Kv7.4^Q71Sfs^, Kv7.4^W242X^, and Kv7.4^A349Pfs^ (Figure 1), in HEK293T-based stable cell lines. Unexpectedly, we encountered great difficulties in establishing stable cell lines that were to constitutively express these variants, especially for Kv7.4^W242X^. After confirming the reproducibility of this issue, we suspected inherent cytotoxicity in these truncated variants. By performing CellTox Green Cytotoxicity and RealTime-Glo Annexin V Apoptosis assays using doxycycline-inducible stable cell lines, we found that these three Kv7.4 variants are indeed cytotoxic and induce cell death to various degrees in a doxycycline-dosage-dependent manner [41]. The cell-death-inducing cytotoxicity of these Kv7.4 variants was further confirmed in an HEI-OC1 cell line that was derived from the murine inner ear [41]. We also found that none of these truncated Kv7.4 variants were functional by themselves (when singly expressed); nor were they capable of forming a heteromer with Kv7.4^WT^ [41], refuting the possibility that Kv7.4^W242X^ and Kv7.4^A349Pfs^ physically interact with Kv7.4^WT^ to exert either an inhibitory (dominant-negative) or cooperative effect.

A CellTox Green Cytotoxicity assay was also performed for HEK293T-based doxycycline-inducible stable cell lines expressing truncated Kv7.1 variants lacking the C-terminal tetramerization domain, Kv7.1^E261X^, Kv7.1^W305X^, Kv7.1^Q530X^, and Kv7.1^Q531X^ (Figure 1), among which solely Kv7.1^Q530X^ was JLNS-associated [115]. Large cell-death-inducing cytotoxicity was found in Kv7.1^E261X^ and Kv7.1^W305X^, while small cytotoxicity was found in Kv7.1^Q530X^ and Kv7.1^Q531X^. A previous study showed that Kv7.1^Q530X^ does not exert a dominant-negative inhibitory effect on wild-type Kv7.1 (Kv7.1^WT^) [5,29]. Consistently, another study showed that Kv7.1^Q530X^ does not bind to Kv7.1^WT^ [31]. It is probable that the other three Kv7.1 variants, which are truncated similarly (Kv7.1^Q531X^) or are shorter (Kv7.1^E261X^ and Kv7.1W^305X^) compared to Kv7.1^Q530X^, are also incapable of physically interacting with Kv7.1^WT^.

Collectively, these observations affirm the inadequacy of a pathological mechanism that is based on dominant-negative inhibition or haploinsufficiency alone for explaining the mode of inheritance of some Kv7 variants.

## 6. The Mechanism Underlying Cell-Death-Inducing Cytotoxicity

Both Kv7.1 and Kv7.4 are membrane proteins that are translated and matured in the ER. It is conceivable that misfolding of truncated Kv7 variants and their accumulation induce chronic cellular stress and eventual cell death. In fact, we detected splicing of XBP1 mRNA, which is indicative of ER stress, in cells expressing Kv7.4 variants [41]. We also found that application of autophagy inducers, imatinib, SB202190, and FK-506, delayed the onset of cell death induced by the three truncated Kv7.4 variants [41], suggesting that accumulation of the truncated Kv7.4 proteins indeed underlies cell-death-inducing cytotoxicity. It should be noted that XBP1 mRNA splicing and small but nonnegligible cell-death-inducing cytotoxicity were also detected in cells expressing Kv7.4^WT^ [41]. The small cytotoxicity of Kv7.4^WT^ may be of physiological relevance because Kv7.4 is abundantly expressed in OHCs and because doxycycline-driven forced overexpression of prestin, another membrane protein abundantly expressed in OHCs, induced much less cell death compared to Kv7.4^WT^ [41]. It may be conceivable that the Kv7.4 protein tends to misfold and self-aggregate even for the wild type, and that the basal endogenous autophagy activity needs to be kept high in OHCs to counteract the cytotoxicity associated with the large expression of Kv7.4. Such a view is in line with a recent study demonstrating that an inhibitor of autophagy, chloroquine, specifically damages OHCs in mice [116]. Pharmacological intervention to maintain or augment the endogenous autophagy activity may delay or prevent age-related hearing loss (presbycusis).

## 7. Updates on the Pathological Mechanisms of JLNS and DFNA2A

A dominant-negative inhibition-based pathological mechanism has been experimentally validated for many dominantly inherited Kv7.1 and Kv7.4 variants. We confirmed that two missense Kv7.4 variants, Kv7.4^G285S^ and Kv7.4^P291L^, can interact with Kv7.4^WT^, and that their cytotoxicity is WT-like [41]. Thus, our study does not challenge the dominant-negative inhibition-based pathological mechanism (at least for these two Kv7.4 missense variants). However, to the best of my knowledge, a haploinsufficiency-based pathological mechanism has never been firmly demonstrated for any Kv7 variants. As mentioned above, a haploinsufficiency-based pathological mechanism is incompatible with several observations and, thus, not compelling.

The cell-death-inducing cytotoxicity identified in our recent study [41] provides a straightforward explanation for the observed dominant inheritance of truncated Kv7 variants lacking the ability to exert a dominant-negative inhibitory effect. It can also explain why some Kv7.1 variants that are associated with severe cardiac phenotypes are not always co-associated with proportionally longer corrected QT (QTc) intervals [117,118,119,120]. For example, a sixteen-year-old proband with Kv7.1^W305X^ died suddenly, and two other affected members experienced syncopal episodes, although the resting QTc interval of patients with this Kv7.1 variant was found to be normal to borderline [120]. The normal-like QTc interval seems reasonable because it is unlikely that Kv7.1^W305X^ exerts a dominant-negative inhibitory effect on Kv7.1^WT^ and because one functional *KCNQ1* allele is sufficient for supporting normal cardiac function (see above). It is conceivable that the severe cardiac phenotypes found in patients with Kv7.1^W305X^ are ascribed to the large cell-death-inducing cytotoxicity found in this variant [41]. The Kv7.1^WT^-like small cytotoxicity found in recessively inherited Kv7.1^Q530X^ among four truncated Kv7.1 variants tested [41] is compatible with such a view.

The vast majority of Kv7.1 variants are inherited dominantly and are associated with LQTS (RWS), but not with hearing loss. Homozygous or compound heterozygous Kv7.1 mutations that are not associated with hearing loss are also reported [5,6]. These observations suggest that the cardiac function is more sensitive to the reduction of Kv7.1 function compared to the auditory function. Thus, as it has been presumed, JLNS hearing loss is probably caused by drastic or complete loss of Kv7.1-mediated K^+^ channel activity. Pathological contributions of cytotoxicity of Kv7.1 variants are likely minimal in JLNS, given the observed recessive inheritance of JLNS variants.

## 8. Remaining Questions and Future Directions

It seems reasonable to speculate that the severity of disease phenotypes and the mode of inheritance of truncated Kv7 variants lacking their C-terminal tetramerization regions is largely determined by the magnitude of their anticipated cytotoxicity. However, the large cytotoxicity found in Kv7.4^A349Pfs^ contradicts the fact that this variant is inherited recessively [92]. Future studies should address if the truncated Kv7 variants are actually expressed in natural host cells. In addition, the vulnerability of natural host cells to cytotoxic Kv7 variants may be very different from those of HEK293T and other cell lines. Animal models need to be generated to fully examine the pathophysiological role of cytotoxic Kv7 variants found in our in vitro study. Animal models are also needed to examine potential adverse effects of drugs (e.g., autophagy inducers). For example, imatinib was found to delay the onset of cell death induced by cytotoxic truncated Kv7.4 variants [41]; however, it was reported to be ototoxic [121,122]. We are currently in the process of generating knock-in mouse models for several truncating *Kcnq4* variants to address these questions.

Despite its limitations, in vitro characterization of disease-associated Kv7 variants using doxycycline-inducible stable cell lines, as showcased in our recent study for Kv7.1 and Kv7.4 [41], will remain useful as an experimental platform for rapidly identifying cytotoxic variants and screening for potentially effective drugs. In general, a dominant-negative inhibitory effect is suspected for dominantly inherited missense Kv7 variants that retain the C-terminal tetramerization region. For example, for Kv7.4 missense variants, a dominant-negative inhibitory effect has been demonstrated for p.L47P, p.N264S, p.S269F, p.S273A, p.L274H, p.W276S, p.T278A, p.L281M, p.L281S, p.G285C, p.G285S, p.L295P, p.G296S, p.G321S, and p.R433W [52,54,58,68,73,74,76]. However, these studies would not rule out the possibility that some missense variants are also cytotoxic. Not a Kv7.4 variant, but the expression of a missense Kv7.2 variant, Kv7.2^M518V^, was reported to induce neuronal death [123]. Missense mutations that were reported to impair cell membrane targeting of Kv7.4, i.e., p.L274H, p.W276S, p.L281S, p.G285C, p.G285S, p.G296S, and p.G321S [58,68,81], are of particular interest because impaired cell membrane targeting implies a structural defect in a mutated protein, an intracellular accumulation of which could induce cellular stress. It should be pointed out that many previous studies reporting a dominant-negative inhibitory effect for Kv7 variants used transiently transfected cells. It is possible that a dominant-negative effect may be erroneously judged for some cytotoxic Kv7 variants that may indirectly affect the overall Kv7 channel activity by impairing cell viability, rather than directly affecting the function of Kv7^WT^. In fact, our whole-cell patch clamp recordings found “apparent” dominant-negative inhibitory effects in cells expressing both Kv7.4^WT^ and Kv7.4^242X^ or Kv7.4^A349Pfs^, which was most likely attributable to impaired cell membrane integrity [41]. Hence, cytotoxicity should be suspected and explored in any variants that affect the amino acid sequence of the Kv7 proteins (Table 1 and Table 2). The use of inducible stable cell lines is strongly encouraged to confidently identify cytotoxic variants and to quantify their relative cytotoxicity.

Since loss of function likely accounts for the pathological role of Kv7.1 variants underlying recessively inherited JLNS, it may seem insignificant to explore cytotoxicity in JLNS-associated Kv7.1 variants. In fact, our recent study identified WT-like small cytotoxicity in a JLNS-associated Kv7.1 variant, Kv7.1^Q530X^ [41]. However, the generality of this finding should be thoroughly examined, especially given the observation that a recessively inherited Kv7.4 variant, Kv7.4^A349Pfs^, exhibits counterintuitively large cell-death-inducing cytotoxicity [41]. The relatively large number of JLNS-associated Kv7.1 variants (Table 1) allows for a systematic study examining the anticipated absence of large cytotoxicity that may account for their recessive inheritance.

Due to the remarkable structural similarity between Kv7.1 and Kv7.4 (Figure 1), a missense change of a conserved residue that affects the function of one Kv7 protein is expected to similarly affect the other. The fact that an approximately ten times greater number of disease-associated variants are found in *KCNQ1* compared to *KCNQ4* implies that the cardiac function is even sensitive to very small reductions in Kv7.1 channel activity. If true, why do JLNS-associated Kv7.1 variants, which induce severe cardiac (and hearing) phenotypes when they are homozygous or compound heterozygous, not induce haploinsufficiency-related cardiac phenotypes when heterozygous? In addition to examining cytotoxicity, it would be also worthwhile to learn if JLNS-associated Kv7.1 variants, especially missense ones, are functional by themselves and capable of physically interacting with Kv7.1^WT^ to exert an inhibitory or cooperative effect. These experimental efforts may contribute to the determination of the magnitude of Kv7.1-mediated K^+^ channel activity that is minimally required to support normal cardiac and auditory functions.

It should be noted that “disease-associated” variants are not necessarily pathogenic. Experimental characterization is important for any disease-associated variants to define their pathogenicity. It is entirely possible that some variants have been falsely identified as pathogenic or likely pathogenic. Identification of nonpathogenic variants is as important as identification of truly pathogenic variants. The significance of experimental validation should not be understated regardless of the likelihood of a variant being pathogenic or nonpathogenic.

## 9. Concluding Remarks

Cytotoxicity-based pathological mechanisms have been acknowledged to account for several neurodegenerative diseases, such as Alzheimer’s disease, in which pathological aggregations of the tau protein are recognized as one of the hallmarks of the diseases [124], with over 100 dominantly inherited tau variants reported to date. However, cytotoxicity-based mechanisms have barely, if not never, been entertained for ion-channel-related hereditary hearing loss. Our efforts to experimentally confirm the absence of K^+^ channel activity in truncated Kv7.4 variants lacking the C-terminal tetramerization region might be seen as belaboring, given the fully established fact that tetramer formation is essential for completing a functional channel pore in any Kv channel. In fact, all three truncated Kv7.4 variants examined in our recent study were found to be nonfunctional by themselves, just as expected. However, cell-death-inducing cytotoxicity would have not been found if we did not pursue this thoroughly using inducible stable cell lines, underscoring the importance of experimentally characterizing disease-associated variants without preconceptions.

We identified large cell-death-inducing cytotoxicity in truncated Kv7.1 and Kv7.4 variants. However, as stressed above, cytotoxicity should be suspected in any variants, especially for dominantly inherited ones, that affect amino acid sequences of the Kv7 proteins. The currently pursued pharmacological strategy to augment residual Kv7 channel activity would not benefit patients with cytotoxic Kv7 variants. Thus, cytotoxic variants must be identified and distinguished from the others so that legitimate clinical strategies can be developed separately.

## Figures and Tables

**Figure 1 biomedicines-10-02254-f001:**
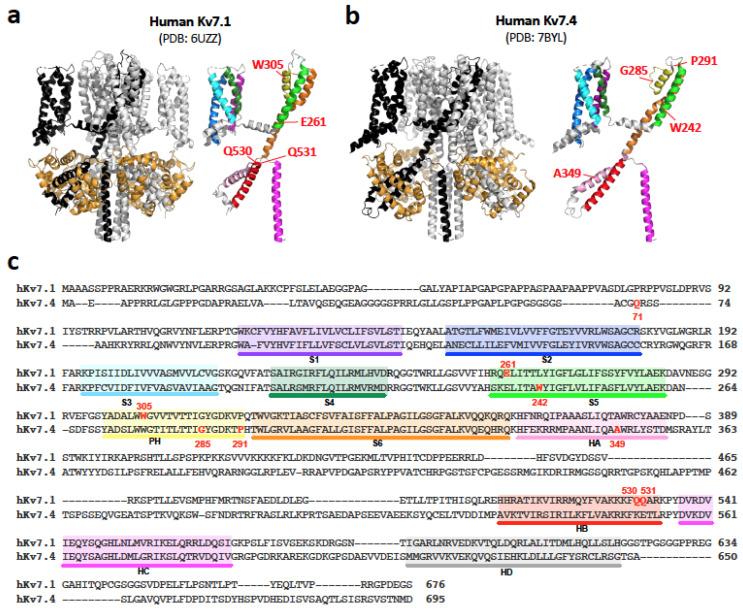
The structures of human Kv7.1 (**a**) and human Kv7.4 (**b**). One of the four protomers and four bound calmodulins (one calmodulin per protomer) are shown in black and pale orange, respectively. In both (**a**) and (**b**), a single protomer is shown on the right. The residues referred to in the main text are shown in red. (**c**) The amino acid sequences of human Kv7.1 and human Kv7.4. The highlights indicate helices, whose colors are matched with those used in (**a**,**b**).

**Table 1 biomedicines-10-02254-t001:** JLNS-associated *KCNQ1* variants that affect the amino acid sequence of the Kv7.1 protein.

Variant *		Functional Study ^†^	
c.2T>C	translation starts from Met159?	[21]	[21]
c.115G>T	p.E39X	[22]	
c.546C>A	p.S182R	[23]	
c.557G>A	p.G186D	[24]	
c.604G>A	p.D202N	[25]	[26]
c.728G>A	p.R243H	[27]	[27,28,29,30,31,32,33]
c.775C>G	p.R259G	[34]	
c.783G>C	p.E261D	[35]	[29,31]
c.815G>A	p.G272D	[35]	
c.914G>C	p.W305S	[36]	[27,37]
c.1040T>G	p.L347R	[38]	
c.1051T>C	p.F351L	[24]	
c.1175G>A	p.W392X	[39]	
c.1588C>T	p.Q530X	[35]	[5,29,31,40,41] ^‡^
c.1741A>T	p.K581X	[42]	
c.431delC	p.I145Sfs	[39]	
c.443delA	p.Y148Lfs	[24]	
c.451_452delCT	p.L151Gfs	[43]	
c.585delG	p.L196Sfs	[25]	
c.733_734delGG	p.G245Rfs	[44]	
c.820_830del11	p.I274Vfs	[45]	
c.998_999delCT	p.S333Cfs	[46]	
c.1008delC	p.I337Sfs	[35]	[29,31]
c.1188delC	p.R397Gfs	[47]	
c.1319delT	p.V440Afs	[48]	
c.1356delG	p.L453Wfs	[49]	
c.567dupG	p.R190Afs	[50]	
c.1149dupT	p.A384Cfs	[21]	[21]
c.743_744delGGinsTC	p.W248F	[51]	[40,51]
c.1630_1635delCAGTACinsGTTGAGA	p.Q544Vfs	[4]	[27,37]

* Multiple case studies are reported for many variants, but only the primary report is provided for each. JLNS-associated *KCNQ1* variants that were also reported as dominantly inherited and LQTS-associated are not included. ^†^ Most functional studies reported significantly reduced or lost K^+^ channel activity and little to small dominant-negative inhibitory effects on Kv7.1^WT^, which account for the recessive inheritance of the JLNS-associated Kv7.1 variants. Two studies [30,33] reported reduced sensitivity to PIP_2_. ^‡^ Our recent study [41] examined the cell-death-inducing cytotoxicity.

**Table 2 biomedicines-10-02254-t002:** DFNA2A-associated *KCNQ4* variants that affect the amino acid sequence of the Kv7.4 protein.

Variant *		Functional Study ^†^	
c.140T>C	p.L47P	[54]	[54]
c.343C>G	p.L115V	[55]	
c.463G>A	p.G155R	[56]	
c.546C>G	p.F182L	[57]	[58]
c.572C>T	p.A191V	[59]	
c.650T>A	p.M217K	[60]	
c.689T>A	p.V230E	[61]	[62]
c.701A>T	p.H234L	[63]	
c.709G>A	p.E237K	[59]	
c.725G>A	p.W242X	[64]	[41] ^‡^
c.754G>C	p.A252P	[56]	
c.767T>G	p.V256G	[59]	
c.770A>G	p.Y257C	[59]	
c.773T>C	p.L258P	[59]	
c.778G>A	p.E260K	[64]	[62]
c.785A>T	p.D262V	[64]	[62]
c.796G>T	p.D266Y	[65]	[65]
c.808T>C	p.Y270H	[66]	[62]
c.821T>A	p.L274H	[67]	[58,68]
c.823T>C	p.W275R	[69]	[62]
c.824G>C	p.W275S	[70]	
c.827G>T	p.W276L	[71]	
c.827G>C	p.W276S	[72]	[58,68,73,74]
c.842T>C	p.L281S	[75]	[58,68]
c.842T>G	p.L281W	[59]	
c.853G>T	p.G285C	[72]	[58,68,73]
c.853G>A	p.G285S	[76]	[52,68,76,77] ^¶^
c.857A>G	p.Y286C	[60]	
c.857A>C	p.Y286S	[78]	
c.859G>C	p.G287R	[79]	[62]
c.872C>T	p.P291L	[61]	[62]
c.871C>T	p.P291S	[61]	[62]
c.878C>T	p.T293I	[59]	
c.887G>A	p.G296D	[80]	
c.886G>A	p.G296S	[81]	[58,68,81]
c.889A>G	p.R297G	[59]	
c.891G>T	p.R297S	[61]	
c.947G>T	p.G316V	[82]	
c.956G>A	p.G319D	[83]	[83]
c.961G>A	p.G321S	[72]	[58,68]
c.992G>A	p.R331Q	[83]	[83]
c.992G>C	p.R331P	[59]	
c.1012C>G	p.R338G	[84]	
c.1012C>T	p.R338W	[59]	
c.1288G>A	p.E430K	[85]	
c.1316G>A	p.R439H	[84]	
c.1365G>T	p.H455Q	[86]	[74,81]
c.1498C>T	p.R500C	[59]	
c.1600A>G	p.I534V	[82]	
c.1647C>G	p.F549L	[87]	
c.1762G>C	p.G588R	[59]	
c.2014G>A	p.V672M	[88]	
c.2039C>T	p.S680F	[89]	[62]
c.211delC	p.Q71Sfs	[90]	[41] ^‡^
c.212_224del13	p.Q71Pfs	[72]	
c.261_269delCTACAACGT	p.Y88_V90del	[65]	[65]
c.664_681del18	p.G222_L227del	[73]	[73]
c.806_808delCCT	p.S269del	[91]	[83]
c.811_816delGCCGAC	p.A271_D272del	[83]	[83]
c.1044_1051delTGCCTGGC	p.A349Pfs	[92]	[41] ^‡^
c.1725delG	p.I576Sfs	[93]	
c.228_229dupGC	p.H77Rfs	[61]	
c.1671_1672dupACGAC	p.V558Tfs	[94]	

* Multiple case studies are reported for many variants, but only the primary report is provided for each. Seven potentially pathogenic *KCNQ4* missense variants (p.N264S, p.S269F, p.S273A, p.T278A, p.L281M, p.L295P, p.R433W) reported in gnomAD and characterized in Jung et al. [74] are not included. ^†^ These functional studies reported significantly reduced or lost K^+^ channel activity (or membrane targeting) and severe dominant-negative inhibitory effects on Kv7.4^WT^, except for p.F182L (Kv7.4^WT^-like), p.H455Q (Kv7.4^WT^-like), and p.G319D (nonfunctional when singly expressed, but gains hyperactivity when co-expressed with Kv7.4^WT^). ^‡^ Our recent study [41] examined the cell-death-inducing cytotoxicity. ^¶^ Functional characterization in a mouse model [52].

## Data Availability

Not applicable.

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
