# Peer review of "The Pathological Mechanisms of Hearing Loss Caused by KCNQ1 and KCNQ4 Variants"

_biomedicines, 2022, doi:10.3390/biomedicines10092254_

Round 1

Reviewer 1 Report

This is a very well written review based upon strong evidence obtained by the author's group and thus convincing.

This review reports a novel and unexpected pathogenesis of hearing loss contributed by KCNQ1 and KCNQ4 genes, which are one of the several dozen genes implicated in hereditary hearing loss.   The author's group in their previous study reported cytotoxic cell death mechanism may implicated in the Kv7.1 and Kv7.4 variants, which are autosomal dominant inheritance, originally thought to be due to dominant negative inhibition or haploinsufficiency. However, the variation in expression which is not rarely seen in autosomal dominant inheritance easily made us to down scale the underlying pathogenesis of such variations. This review reported a very nice demonstration that such variation may caused by other adjunctive or alternative mechanisms.   The author is advised to summarize if such mechanism exist in other hereditary disorders, especially those with AD inheritance, then this review can offer a even broader insight of the complex mechanisms underlying variable expressivity, noted as different severities and different affected organs/systems in the same family or different unrelated families to have the same diseases and thus offer a greater value to medical genetics.

Author Response

Thank you so much for your comments on my review article.

Please see the attachment for my responses.

Reviewer 2 Report

 The review manuscript submitted to „biomedicines” summarises and discusses the role of variants of potassium ion channel proteins for hearing loss, with a specific focus on the pathological mechanism. The topic is an important contribution fitting very well to a Special issue on “Genetic Research on Hearing Loss”, in the broad group of channeloptahies.

Genes for Kv7 channels belonging to 2 variants, KCNQ1 and KCNQ4, relate to different clinically relevant effects and syndromes of hearing loss. While several sites of genetic variation have been identified, the pathogenic mechanism was only recently revealed with cell death.

The manuscript is concise and written in a clear style to assemble disparate information from numerous studies and highlight important pathways for future research to address the mechanisms of hearing loss related to KCNQ variants. The introduction and overall structure of the article are clear and serve the overview and evaluation of physiological effects of mutations in the ion channels. The introduction directs the focus to the Kv7.1 and Kv7.4 channels, following the recent insights into the role of cytotoxicity, with further discussion of the mechanisms in section 6 and 7. Critically, this findings change the understanding of the related hearing loss from models of dominant-negativ inhibition. As pointed out by the author, this also opens a novel avenue of pharmacological research on hearing loss.

The review is an up to date and also timely survey on what is known and what is not known as yet regarding the important roles of voltage-gated potassium ion channels in the occurrence of hearing loss.

I have only few comments on linking the information from different parts:

L. 35   Would RWS also be appropriate as key word?

L. 39/ 70         the role of marginal cells and also structure of the hearing organ are mentioned in different places here. Consider to bring this specific information (L. 66 – 67) into the introduction (L. 42 mentioning also the hair cells) as a short introduction on the hearing basis.

L. 65   Relating to title of the manuscript, consider to add the number of gene variant mentioned in section 2 also in the abstract, which is rather short. This would highlight the importance of the research by high number of variants.

L. 89 Given the high numbers of variants relating to hearing loss, a brief mention would be welcome if actually the Kv7 channels are the most relevant or most numerous group for variants associated with hearing loss – what about other Kv groups, and are there different types of channels also relevant.   

L. 110 variants (spelling)

L. 298 The list of variants in the Tables 1 and 2 in certainly a valuable compilation of references, but in the current form, the tables are somewhat isolated from the remaining manuscript – in fact, they are only referred to twice in the main text (on page 6). They don’t give much further information for themselves, only refer to other publications. For a reader of the review not familiar with the individual studies cited here, it would be helpful to integrate the information further with the manuscript, and also possibly include what specific effect was described for the different gene variants.

Author Response

(The authors gave the same response as above.)
